# Spatial Restrictions in Chemotaxis Signaling Arrays: A Role for Chemoreceptor Flexible Hinges across Bacterial Diversity

**DOI:** 10.3390/ijms20122989

**Published:** 2019-06-19

**Authors:** David Stalla, Narahari Akkaladevi, Tommi A. White, Gerald L. Hazelbauer

**Affiliations:** 1Electron Microscopy Core Facility, W117 Veterinary Medicine Building, 1600 East Rollins St., University of Missouri, Columbia, MO 65211, USA; ds5x8@mail.missouri.edu (D.S.); whiteto@missouri.edu (T.A.W.); 2Department of Biochemistry, 117 Schweitzer Hall, University of Missouri, Columbia, MO 65211, USA; akkaladevin@missouri.edu

**Keywords:** Bacterial chemotaxis, transmembrane receptors, helical bends, molecular crowding

## Abstract

The chemotactic sensory system enables motile bacteria to move toward favorable environments. Throughout bacterial diversity, the chemoreceptors that mediate chemotaxis are clustered into densely packed arrays of signaling complexes. In these arrays, rod-shaped receptors are in close proximity, resulting in limited options for orientations. A recent geometric analysis of these limitations in *Escherichia coli*, using published dimensions and angles, revealed that in this species, straight chemoreceptors would not fit into the available space, but receptors bent at one or both of the recently-documented flexible hinges would fit, albeit over a narrow window of shallow bend angles. We have now expanded our geometric analysis to consider variations in receptor length, orientation and placement, and thus to species in which those parameters are known to be, or might be, different, as well as to the possibility of dynamic variation in those parameters. The results identified significant limitations on the allowed combinations of chemoreceptor dimensions, orientations and placement. For most combinations, these limitations excluded straight chemoreceptors, but allowed receptors bent at a flexible hinge. Thus, our analysis identifies across bacterial diversity a crucial role for chemoreceptor flexible hinges, in accommodating the limitations of molecular crowding in chemotaxis core signaling complexes and their arrays.

## 1. Introduction

The chemotactic sensory system enables motile bacteria to move toward favorable environments [1,2]. The fundamental structural and functional unit of the bacterial chemotaxis system is the core signaling complex [3,4,5]. Core complexes contain two trimers of chemoreceptor homodimers joined by a dimeric histidine kinase CheA, with a coupling protein CheW bound to each kinase protomer (Figure 1A). These core complexes associate into extensive hexagonal arrays by polymerization of the complementary faces of CheW and the P5 domain of CheA [6,7]. In core complexes and their arrays, rod-like chemoreceptors are in close proximity to each other in an environment of molecular crowding. Trimers of chemoreceptor homodimers extend from each CheA-CheW pair in a core complex (Figure 1B), and thus from each hexagonal vertex in an array. Receptor homodimers are generally depicted as straight rods, as they are shown in Figure 1. However, our recent geometric analysis of *Escherichia coli* chemotaxis signaling arrays, using published values for the dimensions and angles, revealed that straight receptor dimers would clash with their receptor neighbors, and thus would not fit into the available space [8]. However, clashes could be avoided by shallow angles of bending at the two receptor flexible hinges we identified in the receptor cytoplasmic domain [8].

One of these hinges, the HAMP hinge, is at the short, two-strand gap between the membrane-proximal, four-helix-bundle HAMP (Histidine kinases, Adenylate cyclases, Methyl accepting proteins and Phosphatases) domain and the membrane-distal, extended four-helix coiled coil of the chemoreceptor cytoplasmic domain. The other, the glycine (Gly) hinge, is in the midst of the extended coiled coil at a cluster of glycines in a region suggested to be a flexible bundle [9,10]. Figure 2A summarizes our previous observations about the geometry of *E. coli* core complexes and arrays, illustrating that straight receptors would collide, but that collisions could be avoided by receptor bending at one or both of the flexible receptor hinges, albeit over a narrow window of shallow angles of bending [8]. Figure 2B illustrates the extensive structural clashes among straight chemoreceptor dimers in a small array of *E. coli* core complexes, and an example of an avoidance of structural clashes by bending at the two flexible hinges.

The current study extends the characterization of geometric restrictions in chemotaxis signaling arrays beyond the parameters defined for *E. coli* [8], to include variations in the dimensions and angles of chemoreceptors, core complexes and their arrays. We observe a predominant pattern of structural clashes among straight receptors that can be avoided by receptor bends at a flexible hinge.

## 2. Results

We used a geometric modeling tool we had developed to characterize spatial restrictions in *E. coli* signaling complexes and their arrays [8], to investigate such restrictions in signaling complexes and arrays with different dimensions and angles. In *E. coli* signaling complexes, straight receptors clash, no matter what their orientation, but structural clashes could be avoided by bends at the two flexible hinges, the HAMP and Gly hinges [8]. Over a limited window of angles, bends at the Gly hinge alone are sufficient to avoid clashes (Figure 2A). To limit the number of variables in our current analyses, we considered bending only at the Gly hinge/flexible bundle. In addition, we fixed the hinge position 75 Å from the membrane-distal tip of the receptor cytoplasmic domain, where it is located in the receptors of *E. coli*, and close to its location relative to the membrane-distal tip of many other receptor species [10].

### 2.1. Geometric Restrictions as A Function of Chemoreceptor Length

Receptor lengths differ among bacterial species, and in some cases among the receptors in the same species [10,11]. How do differences in the receptor length influence the pattern of allowed receptor orientations in the chemotaxis signaling arrays? We investigated using our modeling tool to identify these geometric restrictions as a function of receptor length and bending angle at the Gly hinge/flexible bundle, while holding other dimensions and angles to those defined for *E. coli*. We found that straight receptors clashed, but clashes could be avoided by bends at the Gly hinge (Figure 3).

How do the receptor lengths displayed in Figure 3 relate to the range of chemoreceptor lengths across bacterial diversity? That range is not well defined, because long-axis lengths are reported for only a few receptor species [8,12,13]. As an alternative, we used dimensions for chemoreceptor domains and segments that are available in the literature, in order to estimate the potentially shortest and longest chemoreceptor that could be assembled from the segments of known dimensions. Those components included variable lengths of the coiled coil region of the cytoplasmic domain [10,11], differing dimensions of known structures for chemoreceptor ligand-binding domains [14], variation in the number of 35 Å-long HAMP domains between the transmembrane and the coiled coil segments [15,16], and a 44 Å length of the transmembrane region [8,17]. Combining the respectively lowest or highest values for each segment then generated an estimate of the length of the shortest chemoreceptor as 170 Å, and the longest as 545 Å (see Materials and Methods). Within that range, straight receptors would collide, but allowed orientations could be created by bends over a limited window of angles at the glycine hinge (Figure 3). The window narrowed with increasing receptor length.

### 2.2. Geometric Restrictions as A Function of Dimer Deflection from the Trimer Central Axis

The dimers in a chemoreceptor trimer of dimers splay from the trimer central axis (Figure 1), resulting in structural clashes between straight receptor dimers from the adjacent trimers of dimers in *E. coli* core complexes and their arrays (Figure 2, [8]). The primary source of information about the angle of deflection is the crystal structure for a trimer of a fragment of the cytoplasmic domain of the *E. coli* chemoreceptor Tsr [18]. In this structure, each dimer is deflected 13° from the trimer central axis. In electron tomography of chemoreceptor arrays, the positions of electron densities for chemoreceptors are roughly consistent with this geometry [4,5,19]. However, it is not known to what degree the deflection angle might vary dynamically, or in organisms not yet characterized. Thus, we investigated the influence of deflection on allowed receptor orientations. We used our modeling tool to plot dimer deflection angle versus bending at the Gly hinge/flexible bundle while holding other dimensions and angles to those defined for *E. coli*. The resulting plot identifies geometric restrictions on receptor positioning (Figure 4). There is a narrow window of allowed receptor orientations in which structural clashes between the splayed receptors could be avoided by bending at the glycine hinge. For a limited range of dimer deflections from the trimer axis, centered at ~3°, straight receptors would not clash, but the receptors would be almost parallel. Splaying angles between ~3° and 0° could be accommodated by a negative deflection at the glycine hinge, thus avoiding structural clashes at the wider, membrane-proximal portion of the periplasmic domain. The narrow window of the allowed combination of angles indicates that variation in the trimer deflection angle is greatly limited by molecular crowding.

### 2.3. Structural Clashes as A Function of Trimer Separation

In the hexagonal geometry of chemotaxis arrays, trimers of chemoreceptor dimers are separated by the distance between vertices, measured by electron tomography as ~7.5 nm for *E. coli* [5]. Electron tomography of chemoreceptor arrays in thirteen bacterial species representing diverse bacterial groups revealed a conserved repeat distance between hexagonal centers [11], and thus of distances between vertices. That aforementioned conservation thus implies that the distance between the trimers of receptor dimers in core complexes and their arrays does not vary significantly across bacterial diversity. However, the averaged patterns of electron tomography might conceal significant variations from a predominant geometry. In addition, any species not yet characterized for array geometry might exhibit alternative hexagonal dimensions, for instance as the result of different dimensions of the components of the hexagonal rings, CheW and the P5 domain of CheA, or the substitution of alternative proteins in the baseplate [20]. Thus we investigated the influence of trimer separation on allowed receptor orientations in chemotaxis signaling arrays. We used our modeling tool to plot this trimer separation versus the glycine hinge bending angle, while holding other dimensions and angles to those defined for *E. coli*. The resulting plot identified the combinations of parameters that would be forbidden by structural clashes, and those that would be allowed because clashes were avoided (Figure 5). At the documented trimer separation of ~75 Å [5], receptor clashes would occur for straight receptors, but could be avoided by receptor bending at the Gly hinge. To eliminate any clashes between receptors devoid of bends, trimer separations would have to be over 50% larger (>117 Å) than the apparently conserved separation of 75 Å. Separations less than ~62 Å would generate clashes that could not be corrected by any angle of bending at the glycine hinge.

### 2.4. Avoiding Structural Clashes in the Absence of Receptor Bends

We considered the extent to which any structural clashes among chemoreceptors might be avoided in the absence of any bends at the flexible hinges. Figure 6 shows the pattern generated by a varying chemoreceptor length, a parameter known to be different across bacterial diversity, and in some cases within a single species [10,11], and dimer deflection from the central trimer axis, a parameter that might vary dynamically or across bacterial diversity. Except for the very shortest chemoreceptors, the structural clashes of straight receptors would be avoided only for dimer deflections significantly less than the 13° dimer deflection documented for *E. coli*. For that 13° deflection, straight receptors would clash throughout the estimated range of receptor lengths.

A 75 Å separation between trimers of dimers is widely conserved across bacterial diversity [11]. However, if there are situations, or as-yet-uncharacterized species, in which trimer separation had different values (see the previous section), the combinations of trimer separation and the receptor length (Appendix A), or of trimer separation and dimer deflection (Appendix A) would generate allowed orientations only for values that are significantly different from any known or projected values.

To what degree could structural clashes in core complexes and arrays be avoided by a gradual curving of the extended coiled coil of the chemoreceptor cytoplasmic domain? We investigated by determining both the allowed and the forbidden orientations as a function of receptor curving and receptor length, dimer deflection or trimer separation. As seen in Appendix A, over a limited window of values, receptor curvature in combination with a variation in one of the other core complex parameters could avoid any receptor clashes, and thus substitute for receptor bends. However, intact, functional chemoreceptor dimers do not exhibit gradual curvature, but do exhibit bends at flexible hinges [8,13]. Thus such bends, not receptor curvature, are the best candidates for avoiding any structural clashes among chemoreceptors in signaling arrays.

Chemoreceptor arrays are preferentially located at cell poles [21,22], where membrane curvature is at its greatest. Might membrane curvature contribute to avoiding structural clashes among neighboring chemoreceptors? We found that such contributions would be minor at best. For membrane curvature to contribute, receptor interaction with the lipid bilayer would have to be sufficiently strong to shift the relative positions of neighboring receptor dimers as a function of that curvature. However, the chemoreceptor transmembrane segments do not have tightly bracketing charged residues [17], and thus it is not clear that there would be forces available to shift receptor orientation. If there were such forces, any changes generated by the physiologically relevant membrane curvature in the relative positioning of neighboring receptor dimers in an array would be small in relation to the changes required to avoid structural clashes. For a membrane with a curvature corresponding to the 0.5 μm radius of an *E. coli* cell pole, the periplasmic tips of neighboring receptors in adjacent core complexes would be separated by ≈ 5 Å relative to their positions, in the absence of membrane curvature (Appendix A). However, a separation of > 42 Å would be required to avoid clashes among straight receptors, as seen in our analysis of trimer separation (Figure 5).

## 3. Discussion

Geometric modeling of bacterial chemoreceptors in chemotaxis core complexes and signaling arrays revealed an environment of molecular crowding that restricted chemoreceptor positioning. We found that the restrictions were sufficiently severe, that straight receptors would not fit in the available space over the wide ranges of dimensions and angles that occur, or might occur, across bacterial diversity, or in the course of receptor dynamics. Receptor positioning was restricted, not only for arrays, in which the six neighboring receptor dimers point toward each other and mutually limit orientational options (Figure 2B), but even for the individual core complexes (Figure 2A). This is because pairs of receptor dimers face each other across the interface between the two trimers of dimers, and limit their respective orientations. The geometric restrictions could be accommodated by some shallow angles of bending at the glycine hinge/flexible bundle subdomain, a region in which such bends are documented [8], and one which is postulated to be a general feature of bacterial chemoreceptors [10,23]. We conclude that, across known and projected variations in the dimensional and geometric parameters of core signaling complexes and their arrays, there are significant restrictions on chemoreceptor positioning imposed by molecular crowding, and suggest that such restrictions are accommodated by bends at receptor flexible hinges. Multiple lines of evidence summarized in the following section indicate that flexible hinges are a general feature of chemoreceptors.

### 3.1. Flexible Hinges in Chemoreceptors

Bending at the glycine and HAMP hinges is documented by electron microscopy of intact, lipid-bilayer-inserted, functional chemoreceptors of *E. coli* [8,13]. In addition, bending at the HAMP hinge has been observed by course-grained molecular dynamic analysis [24]. Bends at these hinges exhibit a range of angles, consistent with flexible hinges, not pivot points between two specific conformational states [8,24]. For the glycine hinge, evolutionary genomic analyses of thousands of chemoreceptor sequences from hundreds of bacterial species identify putative conserved flexible bundle subdomains, some of which include clustered glycines, but all of which include opposing helical packing, that is suggested to destabilize the four-helix bundle, and thus to facilitate hinge bending [10,23]. The HAMP hinge in *E. coli* chemoreceptors is at the three-residue-long, two-helix junction between the four-helix bundle of the HAMP domain and the extended four-helix coiled coil of the cytoplasmic domain [8,13,24], a locus expected to be flexible [25]. Across taxonomic diversity, most deduced chemoreceptor sequences include one or more membrane-proximal HAMP domains connected to a membrane-distal extended coiled coil by a short two-helix segment [15,16]. Thus, most chemoreceptors possess the essential feature of a HAMP hinge.

Electron tomographic images of chemotaxis signaling arrays in vivo suggest that chemoreceptors bend even when incorporated in arrays. Specifically, receptor densities are relatively well resolved near the membrane-distal base plate, but poorly resolved between the glycine hinge and the membrane, consistent with the structural heterogeneity generated by variable bend angles at the glycine and HAMP hinges [4,5,19]. Furthermore, subvolume classification and averaging of chemoreceptor densities for in vivo *E. coli* arrays revealed several receptor orientations in the poor-resolution region that could represent variable bending at flexible hinges [5]. Overall, multiple observations indicate that chemoreceptors bend at glycine and HAMP flexible hinges.

### 3.2. Varying Other Geometric Parameters

In this study, we analyzed the effects of varying several, but not all, of the geometric parameters of core signaling complexes and their arrays. However, our geometric modeling tool allows interested scientists to investigate other variables, such as the dimensions of the periplasmic domain and the position or number of flexible hinges. Extrapolating from our observations, any variation in these parameters could alter the shape and the location of the allowed parameter space, but in most cases, this would have only modest effects on its area. Longer or wider periplasmic domains would reduce the parameter space over which any structural clashes would be avoided. Shorter or narrower domains would increase that parameter space. Shifting the position of a flexible hinge would change the combinations that avoided clashes, but would not affect the area of allowed parameter combinations. Increasing the number of flexible hinges would provide more ways in which bends at those hinges could allow receptors to fit into the available space. For instance, besides the HAMP and Gly hinges, receptor dimers might bend at the two-helix “control cable” between the four-helix transmembrane domain and the four-helix HAMP domain [26].

### 3.3. Heterogeneous and Dynamic Receptor Bending

In the analyses described here, each dimer had the same value for the parameter being varied. For instance in Figure 3, at each value along the ordinate, every receptor dimer in the mathematical model of the complex or the array had the same length, and at each value along the abscissa, every dimer had the same deflection at the glycine hinge. Considering uniform populations provided clarity, and allowed a clear illustration of the consequences of varying the respective parameters. Also, assuming a uniformity of parameter value likely reflected the in vivo situation for receptor length and trimer separation. For receptor length, this is because receptors of different lengths segregate into independent arrays [27,28]. For trimer separation this is because the separation is determined by the dimensions of CheW and the CheA P5 domain [4,5,19], and these components are paired with specific receptor(s) [28,29]. The angle of deflection of receptor dimers at the base of the trimer of dimers might be uniform for a receptor population, or it might vary. It would be uniform if the three-dimer packing interface were sufficiently stable that the angles were fixed. It would be variable if the interface were dynamic. Notably, the distances in vivo between dimers in receptor trimers have been observed to change near the level of the HAMP hinge [30,31,32]. Those changes could reflect an altered deflection at the trimer base or at a flexible hinge. Glycine and HAMP hinge deflections are clearly variable, and likely dynamic [8]. Given that variability, receptor dimers in core complexes or arrays are unlikely to bend in a coordinated manner. Besides variability in the magnitude of bends, there is likely to be variation in the bend direction, toward or away from the trimer central axis, and perhaps variability in the plane of bending relative to the central trimer axis. We suggest that individual receptor dimers dynamically explore multiple positional options within the limits imposed by its neighbors, which themselves are exploring positional options. Thus, individual receptor dimers in core complexes and arrays would vary in their hinge bending angle and direction. As a consequence, chemoreceptor bends, and thus the positions of constituent chemoreceptor dimers would be heterogeneous (Figure 7) and dynamic. For isolated core complexes, in which potential inter-trimer collision partners are limited (Figure 2A), this heterogeneity would provide a wider range than is identified in our analyses of allowed bend angles, and thus positions. For arrays, in which each dimer can experience inter-trimer collisions with five neighboring dimers (Figure 2B), receptor position would still be greatly restricted by neighbors.

### 3.4. Other Roles for Flexible Hinges?

The role of chemoreceptor flexible hinges in the structural geometry of chemotaxis core signaling complexes and their arrays does not preclude additional roles for the receptor bends. These could include a role in chemoreceptor conformational signaling [9,10,23]. If this were the case, one would expect a correlation between receptor bends and the signaling state. For the HAMP hinge in intact, lipid-bilayer-inserted *E. coli* chemoreceptor dimers, neither the bend frequency nor the angle changed with the signaling state [8]. For bends at the Gly hinge, which are more challenging to visualize, more data is needed to determine whether there are correlations with the signaling state [8]. Although specific mutational substitutions at the glycines of the Gly hinge perturb receptor function, the effects of those substitutions on receptor bending has not been characterized. Again, additional data is needed. In any case, independent of a possible role in signaling for receptor bends, the results presented in this study indicate that bending at one or more chemoreceptor flexible hinges can play a crucial role in accommodating the environment of molecular crowding in chemoreceptor core signaling complexes and their arrays across bacterial diversity.

## 4. Materials and Methods

### 4.1. Geometric Modeling of Core Signaling Complexes and Arrays

The Mathematica software suite was used as described previously for *E. coli* [8], and as outlined in more detail in the Appendix A to model the geometry of core signaling complexes and their arrays. Our modeling tool is available as a Mathematica Computable Document Format (cdf) file, that can be downloaded from https://doi.org/10.32469/10355/68127 (accessed on: 23 May 2019). It requires an installation of the free Wolfram Player (https://www.wolfram.com/player/ accessed on: 23 May 2019) or the full Mathematica suite. For *E. coli* modeling we used the published values for the key geometric parameters. For our wider analysis, we varied the dimensions and angles of the receptor dimers, trimers of those dimers and core complexes, but maintained the symmetrical, hexagonal organization of the arrays [11,33] and the placement of trimers of receptor homodimers at each hexagonal vertex [4,5,19,33]. Geometric models were generated as Graphics3D [34] objects using Mathematica 11.1, with each dimer and constituent section rendered as an independent, non-interacting object.

Chemoreceptor homodimers were depicted as uniform 25 Å-diameter, three-dimensional rods with a cap in the mushroom shape of a right circular conical frustum (Figure 1B). The rod represented the four-helix cytoplasmic and transmembrane domains as well as the membrane-proximal, four-helix portion of the periplasmic domain. The frustum represented the membrane-distal portion of the periplasmic domain, which is wider than four helices for many common ligand-binding domains [14]. To limit the number of variables in the analyses reported here, the dimensions of the frustum were those of the commonly-occurring 4HB (four-helix bundle) ligand-binding domain [14,15], for which atomic-resolution structures are available for two *E. coli* chemoreceptors [35,36].

The baseplate of CheW and CheA P5 domains was depicted as a hexagonal template with a side length S_tri_ and a hexagon-center-to-hexagon-center distance √3 S_tri_. Receptor trimers of dimers were positioned at each hexagonal vertex (Figure 2B). Pairs of trimers in mirrored orientations represented a core complex. Hexagonal arrays were assembled with the trimers of neighboring complexes separated by the same distance as between the trimers of a core complex. As in our previous analyses [8], spatial restrictions as a function of hinge bending were investigated in the plane defined by the central axis of each receptor rod, and the central axis of the respective trimer, with the hinge of each receptor dimer bent to the same extent. Similarly, other parameters were varied uniformly across the population of the relevant molecules. If the dimensions and the placement of receptor dimers, trimers of dimers and core complexes resulted in structural clashes, then that orientation was identified as forbidden (Figure 2A, left and right-hand panels). Orientations without clashes were identified as allowed (Figure 2A, top and bottom panels). Clashes were defined as described in the Appendix A, and were shown as overlapping receptor volumes in the resulting illustrations of receptor positioning (Figure 2B, left-hand panels).

### 4.2. Estimating the Shortest and Longest Chemoreceptors

In the absence of a significant number of measurements of the overall chemoreceptor length, we estimated the potential range of receptor lengths across bacterial diversity, using values from the literature for the lengths of the four principle receptor segments: The periplasmic, ligand-binding domain, the transmembrane domain, the HAMP region, and the four-helix coiled coil of the cytoplasmic domain. Approximately three-quarters of chemoreceptors identified by bioinformatics have an extra-cytoplasmic, ligand-binding domain [37]. X-ray structures are available for at least one member of each of the ten domain classes, representing over 90% of the receptor sequences that were identified [14]. These structures have long-axis dimensions with a relatively narrow range of 60 to 95 Å. Of course, receptors identified by bioinformatics as lacking an extra-cytoplasmic domain [14,37] have a length of 0 Å for this receptor segment. For the transmembrane region, the hydrophobic portion is 28.5 Å [17] and the hydrophilic region on each side of the membrane would be expected to be ~7.5 Å [38]. Thus, the transmembrane domain would span ~44 Å, consistent with electron micrographic images of an *E. coli* receptor inserted into a Nanodisc-enclosed, native lipid bilayer [8]. For species with different average lengths of membrane lipids and thus membrane thicknesses, this dimension might vary, but only slightly. Thus, in our estimates of the receptor dimensions we used the single value of 44 Å for the span of the transmembrane domain. A HAMP domain is 35 Å long [39,40]. Bioinformatic analysis has indicated that most chemoreceptors contain one or two HAMP domains, but some contain none, and a few more than two copies [15,16]. In our estimates of the overall receptor length, we focused on the range of 0 to 2 HAMP domains per receptor. Within this range, the length of the HAMP region would vary from 0 to 70 Å for many receptors. The four-helix coiled coil common to all chemoreceptors is a series of paired heptad units [10,41], each pair contributing ~10.5 Å to the receptor length. Bioinformatic analyses of chemoreceptor sequences identified protein classes containing 12 to 32 paired heptads, and thus with coiled-coil lengths of 126 to 336 Å [10]. A few rare receptors appear to be even longer than 32 paired heptads, but we did not consider those unusual examples in estimating the potential range of receptor lengths. Based upon the observed lengths of each chemoreceptor segment, the shortest receptor would have no extra-cytoplasmic or HAMP domain, a 44 Å transmembrane domain and a 126 Å coiled-coil segment, for a total length of 170 Å. The shortest receptor with an extra-cytoplasmic domain would be 230 Å. The longest receptor would have a 95 Å extra-cytoplasmic domain, a 44 Å transmembrane domain, two HAMP domains for a total length of 70 Å, and a coiled coil segment of 336 Å, generating a total length of 545 Å.

## 5. Conclusions

The results presented in this study indicate that bending at one or more chemoreceptor flexible hinges can play a crucial role in accommodating the environment of molecular crowding in chemoreceptor core signaling complexes and their arrays across bacterial diversity.

## Figures and Tables

**Figure 1 ijms-20-02989-f001:**
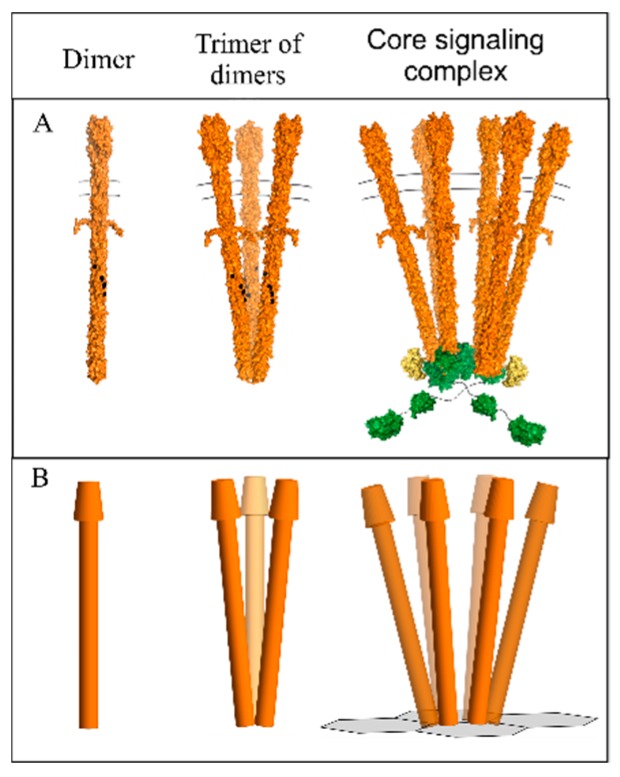
Chemoreceptor organizational units. Models are shown of a chemoreceptor homodimer (left), a trimer of those homodimers (center) and a core signaling complex of two trimers of receptor dimers, the dimeric histidine kinase CheA, and two copies of the coupling protein CheW. (**A**). Space-filling models. (**B**). Representations used in our geometric modeling. In the representation of a core signaling complex, two receptor trimers of dimers are placed at the vertices of the hexagonal grid of the CheW/CheA-P5 domain base plate. See the text for details.

**Figure 2 ijms-20-02989-f002:**
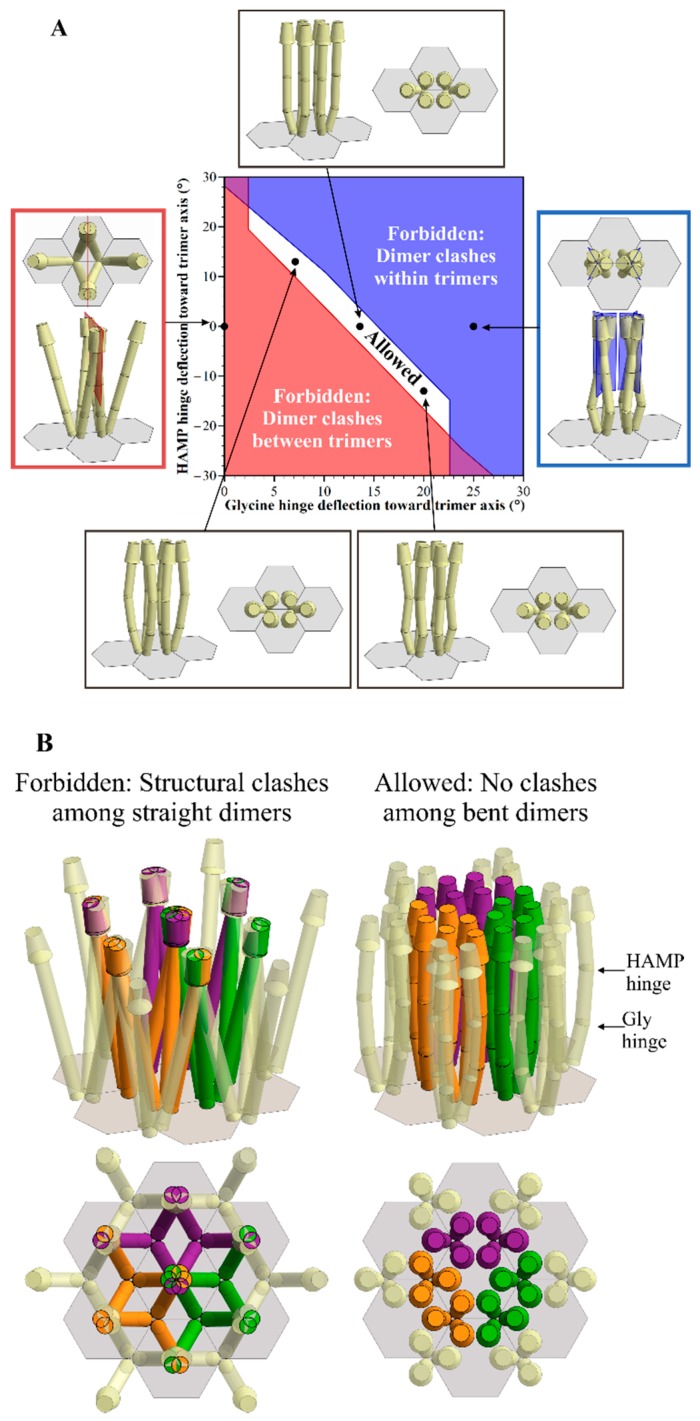
The geometric restrictions in *E. coli* core signaling complexes and their arrays. Dimensions and angles of these core complexes are from the published values as described in Akkaladevi, et al. [8]. Hinge positions are shown as circumferential lines around the receptor rods, the glycine hinge proximal and the HAMP hinge distal to the base plate. (**A**). Some allowed and forbidden combinations of deflection angles at the HAMP and Gly flexible hinges (see the text). The central plot is a modified version of Figure 9 from Akkaladevi, et al. [8], in which white indicates the allowed combinations, red the forbidden combinations because of any clashes between dimers in different trimers, and blue indicates the forbidden combinations because of clashes between dimers in the same trimer. Surrounding the plot are cartoon representations of five specific combinations of hinge deflections for a single core signaling complex within an array, with the respective plot positions indicated by arrows to black dots on the plot. Each representation shows a view parallel to the membrane, and another normal to the membrane from the periplasm. Representations of forbidden orientations show clash planes in blue (within trimers) or red (between trimers). (**B**). A cartoon representation of a twelve trimer-of-receptor-dimers array. The left-hand panels show the forbidden orientation of these receptors with no hinge deflections, i.e., straight receptors, and the right-hand panels portray an allowed orientation of receptors with a 10° deflection of the Gly hinge, and an 8° deflection of the HAMP hinge. The upper panels show views approximately parallel to the membrane, and the lower panels normal to the membrane from the periplasm.

**Figure 3 ijms-20-02989-f003:**
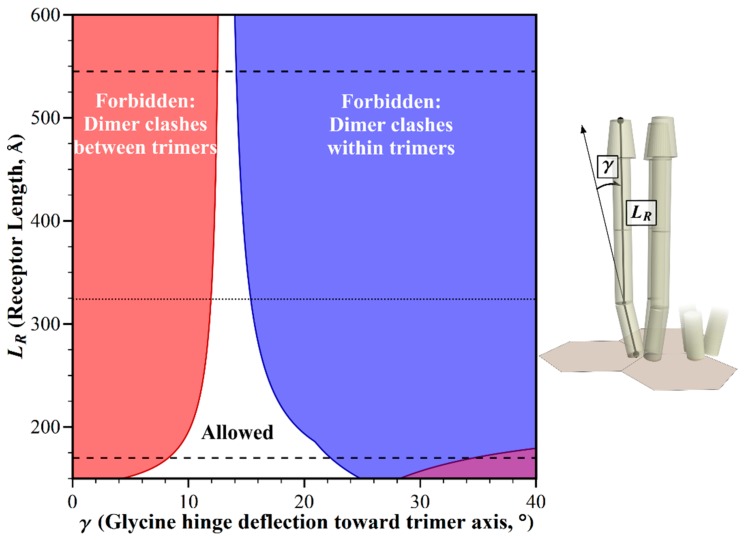
Geometric restrictions in core signaling complexes and their arrays as a function of chemoreceptor length, *L_R_* and glycine hinge deflection, *γ*. The plot shows the structural consequences of the combinations of receptor length and glycine hinge deflection. For allowed combinations (white area), receptors fit into the available space. For any forbidden combinations, they do not, because of structural clashes between dimers in neighboring trimers (red areas), or in the same trimer (blue areas). Estimated lengths of the longest and shortest receptors that could be assembled by combining the longest or shortest known receptor segments (see Materials and Methods), are marked by dashed lines. A dotted line marks the length of the *E. coli* receptors.

**Figure 4 ijms-20-02989-f004:**
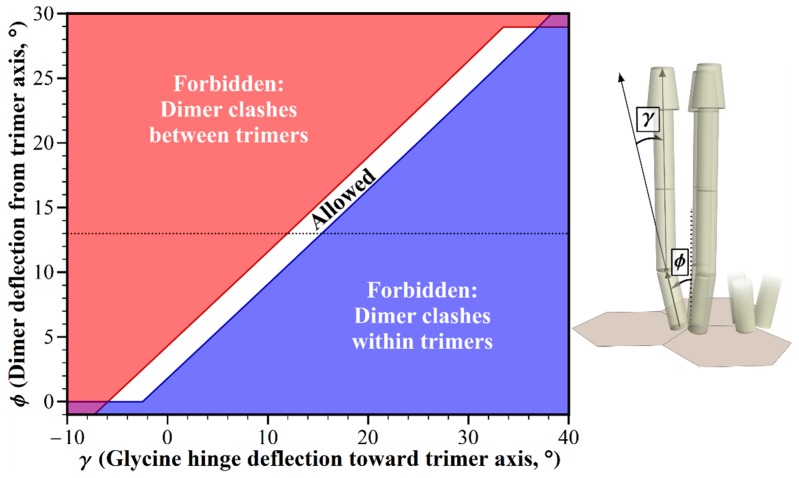
Geometric restrictions in core signaling complexes and their arrays as a function of dimer deflection, *ϕ* and glycine hinge deflection, *γ*. The plot shows the structural consequences of the combinations of dimer deflection from the trimer central axis, and the glycine hinge deflection toward the trimer axis. For the allowed combinations (white area), receptors fit into the available space. For any forbidden combinations, they do not, because of structural clashes between the dimers in neighboring trimers (red areas), or in the same trimer (blue areas). The 13° dimer deflection for an *E. coli* receptor trimer [18] is marked with a dotted line.

**Figure 5 ijms-20-02989-f005:**
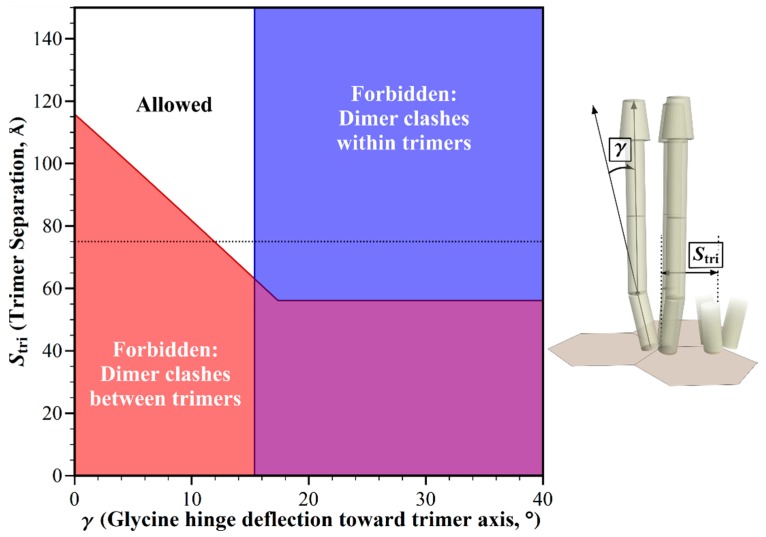
Geometric restrictions in core signaling complexes and their arrays as a function of trimer separation, *S_tri_* and glycine hinge deflection, *γ*. The plot shows the structural consequences of the combinations of trimer separation and glycine hinge deflection. For any allowed combinations (white area), the receptors fit into the available space. For those forbidden combinations, they do not, because of structural clashes between dimers in neighboring trimers (red areas), or in the same trimer (blue areas). The 75 Å trimer separation in *E. coli* chemoreceptor arrays [5] is marked with a dotted line.

**Figure 6 ijms-20-02989-f006:**
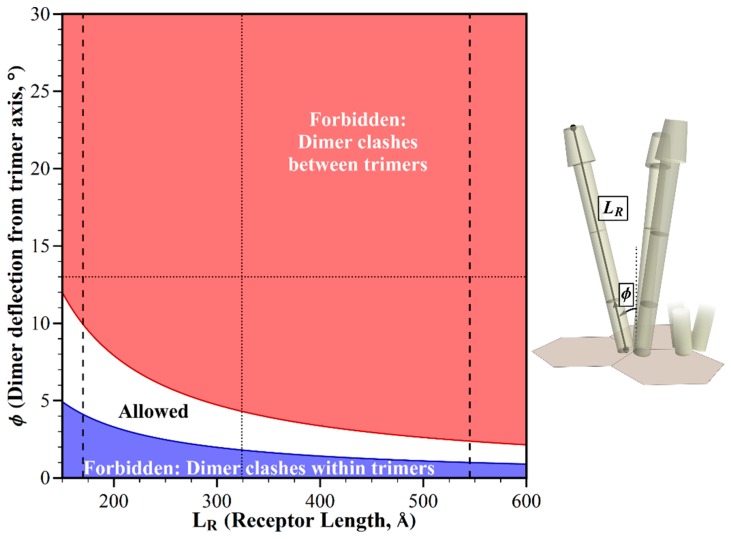
Geometric restrictions in core signaling complexes and their arrays as a function of dimer deflection, *ϕ* and receptor length, *L_R_*. The plot shows the structural consequences of combinations of dimer deflection relative to the trimer central axis and the receptor length. For any allowed combinations (white area), the receptors fit into the available space. For the forbidden combinations, they do not, because of the structural clashes between dimers in neighboring trimers (red areas), or in the same trimer (blue areas). Dashed lines show the postulated shortest and longest chemoreceptor length, as in Figure 3. Dotted lines show respective values for *E. coli* receptors.

**Figure 7 ijms-20-02989-f007:**
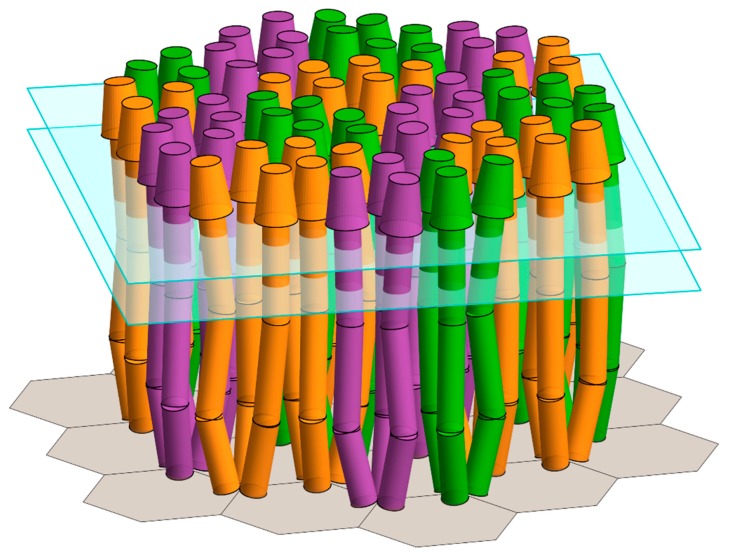
Model of an array of core signaling complexes with heterologous chemoreceptor deflection angles at the HAMP and Gly hinges. The model is an expanded version of the upper right-hand panel in Figure 2B. As in that panel, the dimensions and the angles are those for *E. coli*, and bent chemoreceptor dimers fit into the available space. Different from that panel, the deflection angles at each of the two flexible hinges for each receptor dimer are variable.

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
