# Peer review of "Spatial Restrictions in Chemotaxis Signaling Arrays: A Role for Chemoreceptor Flexible Hinges across Bacterial Diversity"

_ijms, 2019, doi:10.3390/ijms20122989_

Round 1

Reviewer 1 Report

This manuscript explores how the configuration of bacterial chemoreceptors affects their packing in sensory arrays. The main result is that this tight packing tightly constrains the bending/configuration of these receptors. In these regards, structural info can be inferred from packing arguments. Overall, the work is well done. I have only a few minor comments for improvement.

Many geometrical variables are explored in this work. It might help the reader to provide a simple diagram (in the main text) explain to explain exactly what these variables are -- I often struggled trying to decipher them. For example, "dimer deflection from trimer axis" and so on. A simplified version of Figure S3 in the main text, for example, would be extremely helpful.

I was surprised that membrane curvature has such a little effect. This goes against my naive intuition. I would perhaps further emphasize this result.

The results concerning heterogeneity are very interesting. I kept asking this question when I read initially read the manuscript and was pleased to see it discussed. Perhaps you could expand this by bringing in some results from packing theory and statistical physics (this problem has been explored extensively).

Along these lines, you may consider using monte carlo to explore how heterogeneity influences packing. You could sample different configurations using a metropolis-like sample procedure and then show the resulting distributions. Or alternatively, narrowly sample outside of the allowed region. Something to consider for this paper or, perhaps, the next one. 

Author Response

Dear Reviewer,

We have considered carefully your comments about our manuscript “Spatial restrictions in chemotaxis signaling arrays: a role for chemoreceptor flexible hinges across bacterial diversity”.  Our responses to each of those comments are attached here and, as indicated, incorporated into the revised version of our manuscript.

Reviewer 1 comments:

1. “Many geometrical variables are explored in this work. It might help the reader to provide a simple diagram (in the main text) to explain exactly what these variables are…”. We agree that such diagrams would help the reader. Thus we have added a simple cartoon diagram that indicates the respective, relevant variables to accompany the plots in Figs 3, 4, 5, 6, S1 and S2.

2. “I was surprised that membrane curvature has such a little effect. This goes against my naïve intuition. I would perhaps further emphasize this result.” We have expanded the text in the final paragraph of the Results, beginning on lines 218-230, to describe more fully our analysis of the minor effects created by membrane curvature. In parallel, we made some revisions to the legend for Fig. S4.

3. and 4.The reviewer suggests that we might expand our consideration of heterogeneity of receptor bending. We agree that this would be interesting. However, we also agree with the reviewer’s statement in the final sentence of comment 4 that such an expansion is something to consider for the next paper. Thus we have not expanded our consideration of this subject in the current manuscript.

Sicerely

Gerald Hazelbauer

Reviewer 2 Report

the manuscript is well presented and graficaly clear.It is an study of  the chemoreceptors structure and how it  determines the disposition of them same on bacteria surface. The authors have not made any approach to functionality of chemotaxis, even though this aspect would be very interesting.  

Author Response

Dear Reviewer,

We have considered carefully the comment from the second reviewer about our manuscript “Spatial restrictions in chemotaxis signaling arrays: a role for chemoreceptor flexible hinges across bacterial diversity”.  Our response to that comment is attached here and, as indicated, incorporated into the revised version of our manuscript.

Reviewer 2 comments:

1. “The authors have not made any approach to functionality of chemotaxis…”. We thank the reviewer for noting that we had neglected to state that the function of bacterial chemotaxis is to mediate the movement of essentially all motile bacteria toward favorable environments and away from unfavorable ones. In response, we added a sentence at the beginning of the Abstract (beginning on line 12) and the beginning of the Introduction (corresponding to line 29 of the original submitted version).

Sincerely

Gerald Hazelbauer